# Spatial inequalities in life expectancy within postindustrial regions of Europe: a cross-sectional observational study

Martin Taulbut,[1] David Walsh,[2] Gerry McCartney,[1] Sophie Parcell,[2] Anja Hartmann,[3] Gilles Poirier,[4] Dana Strniskova,[5] Phil Hanlon[6]

▶ Prepublication history and additional material is available. To view please visit the journal (http://dx.doi.org/10.1136/bmjopen-2013-004711).

[1]NHS Health Scotland, Glasgow, UK
[2]Glasgow Centre for Population Health, Glasgow, UK
[3]Ruhr-Universität Bochum, Bochum, Germany
[4]Observatoire Régional de la Santé (ORS), Nord-Pas-de-Calais, Loos, France
[5]Regional Public Health Authority of the Olomouc Region of the Czech Republic, Olomouc, Czech Republic
[6]University of Glasgow, Glasgow, UK

**Correspondence to**
Dr Martin Taulbut;
martintaulbut@nhs.net

## ABSTRACT

**Objectives:** To compare spatial inequalities in life expectancy (LE) in West Central Scotland (WCS) with nine other postindustrial European regions.

**Design:** A cross-sectional observational study.

**Setting:** WCS and nine other postindustrial regions across Europe.

**Participants:** Data for WCS and nine other comparably deindustrialised European regions were analysed. Male and female LEs at birth were obtained or calculated for the mid-2000s for 160 districts within selected regions. Districts were stratified into two groups: small (populations of between 141 000 and 185 000 people) and large (populations between 224 000 and 352 000). The range and IQR in LE were used to describe within-region disparities.

**Results:** In small districts, the male LE range was widest in WCS and Merseyside, while the IQR was widest in WCS and Northern Ireland. For women, the LE range was widest in WCS, though the IQR was widest in Northern Ireland and Merseyside. In large districts, the range and IQR in LE was widest in WCS and Wallonia for both sexes.

**Conclusions:** Subregional spatial inequalities in LE in WCS are wide compared with other postindustrial mainland European regions, especially for men. Future research could explore the contribution of economic, social and political factors in reducing these inequalities.

## Strengths and limitations of this study

- This is an extensive international comparison of contemporary, within-region disparities in life expectancy. It compares 100 small districts and 60 large districts across 10 European regions.
- Ecological bias was mitigated by selecting regions with a similar history of deindustrialisation and comparing districts with similar-sized populations.
- While the approach taken here partly addressed the scale issue associated with the 'modifiable area unit problem', it was unable to resolve the zoning issue.
- The study was unable to say whether more heterogeneous populations or higher levels of social segregation were driving these differences, though the limited evidence we have does not support this view.
- The analyses are restricted to one period during the mid-to-late 2000s.
- The approach was restricted to describing spatial differences in life expectancy—we cannot draw any conclusions on within-region inequalities by socioeconomic status, rurality or ethnicity.

## INTRODUCTION

Reducing inequalities in health has been identified as a priority by governments across Europe.[1 2] While inequalities in health are often described using individual characteristics (eg, socioeconomic class), there is also considerable interest in spatial disparities in health,[3 4] despite a lack of research found by Tyner.[5] All countries exhibit subnational variation in mortality and life expectancy (LE).[6–8] The pattern is observed for countries as diverse as France,[9] Sweden,[10] Australia[11] and Poland.[12] Almost universally, the geographical gap in these health outcomes is wider for men than women.[13] There are some observed differences in within-country dispersion in LE, with the spatial gap being more pronounced for some nations (eg, USA[14] and UK[15]) than others (eg, Germany[16] and Poland[12]). Regional inequalities in mortality between English regions, for instance, have been found to be severe and persistent over a 40-year period.[17] Differences are also observed in whether spatial inequality in mortality has been narrowing, static or increasing over time.[13 18] Although the findings are dependent on the size of geographies selected for analysis,[19] there is evidence that inequalities between and within English regions have increased over time.[17 20]

Deindustrialisation has been proposed as a mechanism to partly explain these spatial inequalities. Across Europe, there is a clear

overlap between former coal mining and industrial areas and districts and regions with the poorest health.[7 21] Riva and Curtis[22] found that areas in England with persistently low or deteriorating employment rates (relative to the national average), often located in ex-industrial regions, had the highest rates of mortality and physical morbidity, even after adjusting for migration and individual characteristics of residents. A number of mechanisms (eg, greater poverty, loss of purpose and status and higher levels of substance misuse) provide plausible links between economic dislocation and health outcomes.[23 24]

Making spatial comparisons of health within and between geographies is subject to a number of difficulties. Comparing geographies that have been 'clustered' according to some shared characteristics (such as a similar economic and social history) can partly adjust for this and produce more meaningful results.[25] Geographical comparisons are more valid when the spatial units being compared are of a similar population size and where there is less social diversity within them, since the differences between areas will depend on the degree to which the geographical units of analysis are internally diverse or homogeneous. Units of analysis with larger population sizes or more heterogeneity in their composition are less likely to display differences between areas because of the averaging effect of this greater internal diversity.[19 26] Failing to take this into account may result in misleading comparisons.

The present study approaches this issue from a Scottish perspective. Scotland's position as the 'sick man' of Europe—characterised by a slower rate of improvement in LE compared with other West European nations since the 1950s, and a consequent relative deterioration in its international position—has been discussed elsewhere.[27 28] Furthermore, the within-region spatial gap in mortality was greater in Scotland than any other region of Britain.[29] A similar 'faltering' in the pace of improvement in mortality and LE has also been noted for West Central Scotland (WCS), the region of Scotland most affected by deindustrialisation in recent decades, relative to other postindustrial regions.[30] Postindustrial regions are extremely important in epidemiological terms as they tend to exhibit the highest rates of mortality in their parent countries.[31 32] A recent study also suggested that WCS was more spatially divided in terms of mortality than other comparable European postindustrial regions, though the authors did not pursue this question in depth.[31] This paper explores this question in a systematic way, to investigate whether spatial disparities in mortality within WCS are large compared with other European regions, taking industrial heritage and differences in population sizes of subregions into account.

## METHODS
This study was informed by the authors' involvement in a larger project which aimed to contribute to an understanding of the poor health observed in one postindustrial region, WCS, in the context of other comparable European regions. WCS is a region of 2.1 million people, centred on the City of Glasgow. Nine other regions, highlighted in other recent epidemiological analyses,[30 32] were selected for comparison with WCS. The regions were chosen through consultation with experts on European history on the basis of their shared historic economic dependence on industries such as coal, steel, shipbuilding and textiles, alongside analysis of their subsequent loss of industrial employment over the past 30–40 years.[30]

Table 1 presents summary information on the list of regions selected. Selecting a range of regions from across East and West Europe allowed contrasts to be made between WCS and European areas with different social and political contexts. The inclusion of UK regions meant that WCS could be compared with areas subject to the same set of socioeconomic policies over the past 30–40 years.

Male and female LEs at birth were obtained from relevant statistical agencies (or where appropriate calculated) for the mid-2000s, for 160 districts within the 10 selected regions. Ideally, the years of the data collected would be of identical time frame and size. It was not possible or practical to do so here, because of variation between countries in terms of availability of the required small-area statistics data. All life tables were constructed in the same way, using all deaths within each district and the resident population of each district. The sources of the LE data for each region are given in table S2 (web only table).

In order to reduce the risk of bias due to differing subregional population sizes (the scale problem), we stratified the regions into two. Five regions (Swansea and South Wales Coalfields, Northern Ireland, Nord-Pas-de-Calais, Silesia and Merseyside) had subregional (or district) populations of between 141 000 and 185 000 people. These areas were compared with similarly sized geographies in WCS Community Health Partnership areas (CHPs).[i] Three regions (the Ruhr, Saxony and Wallonia) had LE data calculated across 45 'large' districts of population size ranging from 224 000 and 352 000: these were compared with similarly sized WCS Nomenclature of Units for Territorial Statistics (NUTS) 3 areas. Data for Northern Moravia and WCS were available for both strata. For four regions (Northern Ireland, Wallonia, Silesia and Nord-Pas-de-Calais), it was necessary to create pseudodistricts to ensure a more even distribution of population across districts. This process took into account contiguous boundaries and, where possible, the character of districts. LE at birth was then calculated for these new areas using the Chiang[33] method (II), using population and mortality data obtained from the relevant national statistical agencies.

---

[i]There were 15 CHP areas in WCS prior to April 2010, when the five Glasgow CHPs were merged into three.

**Table 1** Postindustrial regions used in the study, by location, characteristics and population of districts

| Region name | Nation state | Number of districts | Mean population size of districts | Principal historical industries | Total industrial employment loss* |
|---|---|---|---|---|---|
| West Central Scotland | UK | 15† (7)‡ | 141 268† (302 084)‡ | Shipbuilding and support industries (iron, coal, engineering) | −62% (1971–2005) |
| Northern Ireland | UK | 12 | 147 900 | Shipbuilding, textiles, manufacturing | −20% (1971–2005) |
| Merseyside | UK | 9 | 149 532 | Shipping, docks, manufacturing (eg, cement), engineering | −63% (1971–2005) |
| Swansea and South Wales Coalfields | UK | 7 | 160 486 | Coal | −51% (1971–2005) |
| Nord-Pas-de-Calais | France | 25 | 160 746 | Coal, textiles, steel | −43% (1970–2005) |
| Wallonia | Belgium | 11 | 309 542 | Mining, metal working, textiles | −39% (1970–2005) |
| The Ruhr | Germany | 15 | 351 912 | Coal, iron, steel | −54% (1970–2005) |
| Saxony | Germany | 19 | 224 934 | Steel, construction, engineering, textiles | −47% (1991–2005) |
| Northern Moravia | Czech Republic | 11§ | 185 099 | Coal, steel | −19% (1993–2005) |
| Silesia¶ | Poland | 29 | 159 858 | Coal, steel, automobiles, zinc | −55% (1980–2005) |

*Percentage decrease in the number of industrial jobs in each region over the time period shown in parentheses.
For Silesia, change is shown for the Katowice subregion.
†Community health partnerships.
‡Nomenclature of Units for Territorial Statistics (NUTS) 3.
§Jesenik district included in small district comparisons only.
¶Known as the Slaskie region in Poland.

Within regions, we then ranked the subregional (district) populations by their LE separately for men and women and separately for the large and small subregional populations. We then created line graphs for each strata of regions to show the size and distribution of subregional populations and their corresponding LEs. Taking each region separately, we then calculated the range in LE and IQRs, accounting for the population sizes in each subregional district, to describe the within-regional disparities.

## RESULTS
### Regions with small district data (populations between 141 000 and 185 000)
The districts with the highest male LEs (>77 years at birth) were in the rural districts in Northern Ireland, plus the more affluent WCS districts of East Renfrewshire and East Dunbartonshire. The lowest male LEs (<70 years at birth) were in Silesia and in areas of WCS (North and East Glasgow). The districts with the highest levels of female LE (>82.5 years at birth) were all located in Nord-Pas-de Calais, while the districts with the lowest levels of female LE (<78 years at birth) were in WCS (all five Glasgow districts), Merseyside (City and North Liverpool) and parts of the Silesia region (Ruda Slaska-Swietochlowice and Chorzow-Siemianowice Slaskie).

Within regions, the *range* in male LE was widest for WCS (8.6 years) and Merseyside (5.9 years) and narrowest in Swansea and the South Wales Coalfields (1.6 years) and Northern Moravia (2.7 years). The *IQR* in LE for men was widest in WCS and Northern Ireland

(2.7 and 2.6 years, respectively), followed by Silesia (2.2 years), and was much less pronounced in the other regions. For women, WCS had the widest *range* in LE (6.5 years) and Northern Moravia the narrowest (1.6 years). The range of LEs observed for Merseyside districts was also high (5.9). The *IQR* in female LE was highest in Northern Ireland (2 years) and Merseyside (1.9 years) and lowest in Northern Moravia (figure 1).

### Regions with large district data (populations between 224 000 and 352 000)
The highest male LEs were found in Saxony, Wallonia and the Ruhr, while the lowest were observed in WCS (Glasgow), Wallonia (Mons) and in Northern Moravia. For women, districts with the highest LE were located in Wallonia and Saxony, while the districts with the lowest LE were found within WCS and Northern Moravia.

Within regions, the *range* in male LE across 'large' districts was widest for WCS (5.3 years), followed by Wallonia (4.8 years), with the Ruhr Valley, Saxony and Northern Moravia less polarised. The *IQR* in LE was much wider in WCS (3.9 years) than in all other regions. For women, the pattern was similar, with the widest range in LE observed for WCS (3.5 years) and Wallonia (2.5 years), with much less disparity evident in the German and Czech regions (figure 2).

## DISCUSSION
Similarly deindustrialised regions in Europe, which share similar economic, social and health problems,[30 32] display different patterns in spatial inequalities in LE. In

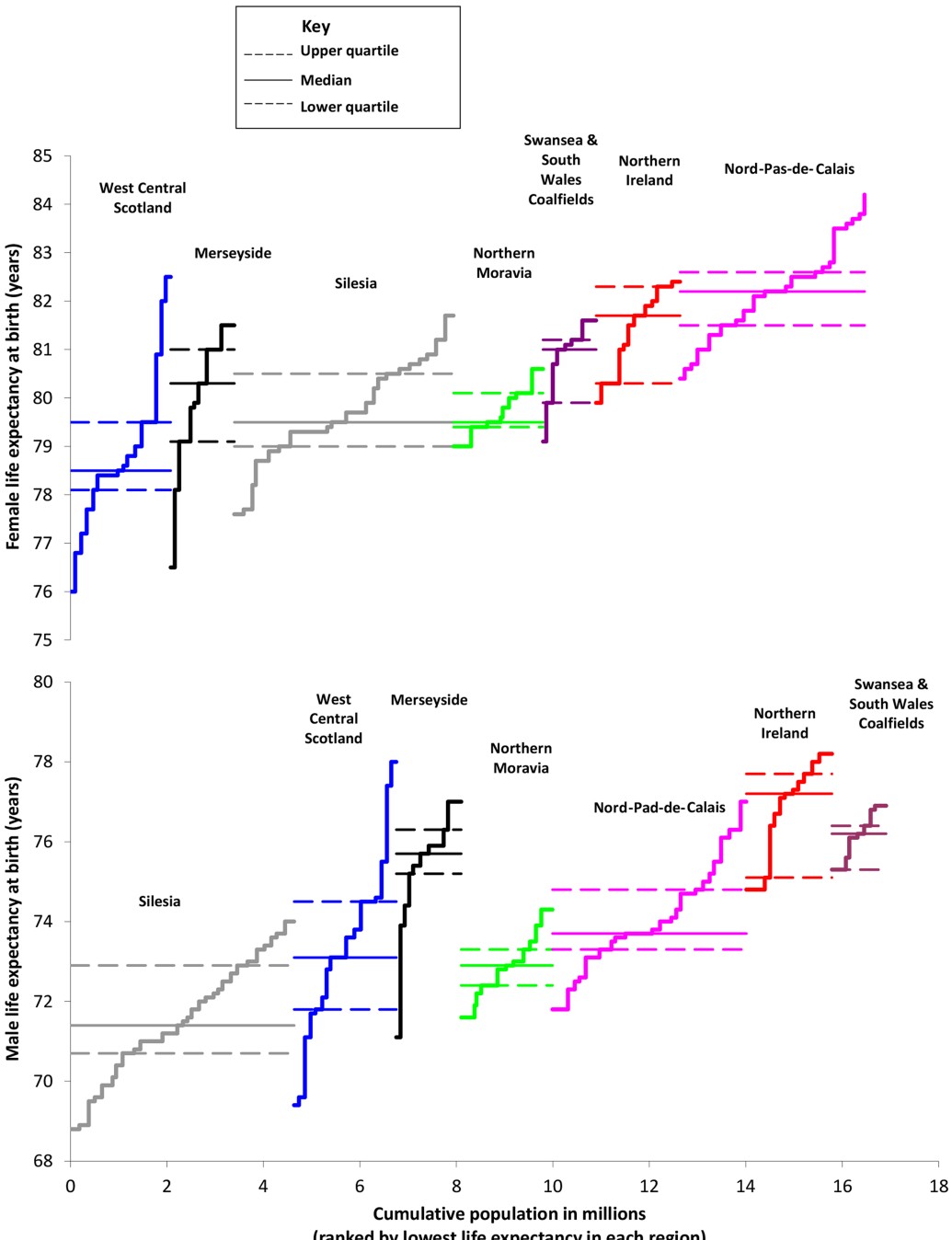

**Figure 1**  IQR of life expectancy for small districts within seven postindustrial European regions, by gender, mid-to-late 2000s.

particular, two UK regions (WCS and Merseyside) have much larger intraregional differences in LE for men and women than the other regions, with WCS having the largest differences. In contrast, there are relatively narrow spatial inequalities in LE in Northern Moravia, the Ruhr and Swansea and South Wales Coalfields.

The present study has four important strengths. First, it provides an original comparison of contemporary, international and within-region disparities in LE. Second, its geographical coverage is extensive: more than 100 small districts and 60 large districts, spanning 10 regions across Western and Eastern Europe. Third, it

uses a straightforward metric of health outcomes (LE at birth) that is readily understood. Finally, by attempting to ensure that the areas are of a similar size and have a common experience of industrial development and subsequent deindustrialisation, the potential bias arising from comparisons of differently sized populations and the heterogeneity within regions is reduced.

The study also has a number of limitations. A key challenge in any study of this kind is the 'modifiable area unit problem' (MAUP). As discussed by Openshaw,[34] the spatial units that can be used to describe individual-level data are usually highly modifiable and their

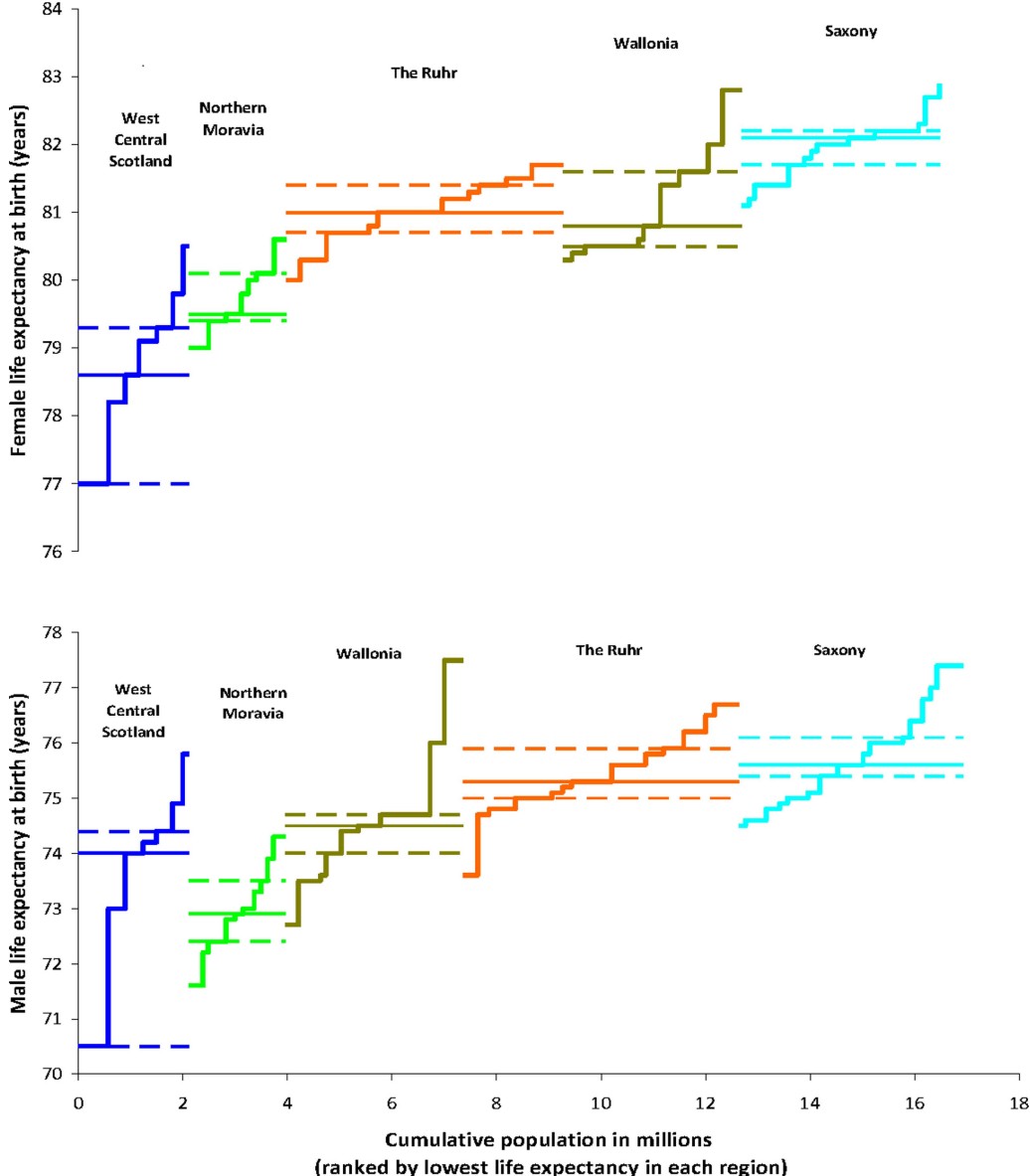

**Figure 2** IQR of life expectancy for large districts within five postindustrial European regions, by gender, mid-to-late 2000s.

boundaries are often decided on an arbitrary basis. There are a large number of different spatial units that could be used to describe the same data, often producing quite different conclusions. There are two components of the MAUP. First, there is a scale problem, with different results being produced depending on the number of spatial units used in analysis (eg, for census tracts, districts, regions). Second, there is a grouping or zoning problem, reflecting different choices about how very small areas are joined together to create areas of a similar size. In this study, the scale problem has been partly addressed by making comparisons of subregional inequalities at two different geographical levels. The similar findings (of greater spatial inequalities in WCS) for both scales can give more confidence that the approach adopted is reasonable. However, the zoning problem remains difficult to resolve without access to

individual-level data coded to geographic areas, which are currently not available. It is important to note that the findings may not apply beyond the selection of post-industrial regions shown here. For example, Hoffman et al,[35] who analysed neighbourhood-level differences in mortality for 15 large European large cities, found that inequalities were wider for women than for men, and there was no evidence that within-area inequalities varied between cities.

The methods used to compare spatial inequalities (IQR) could also be criticised as not ideal. Other studies[36] have used the slope index of inequality and relative index of inequality to estimate spatial inequalities in mortality.[37] This would undoubtedly allow for more robust analyses. However, to allow these indices to be constructed would require robust, internationally comparable measures for ranking all the districts by

socioeconomic status. Data limitations make this a difficult task. Europe-wide indicators of material and income deprivation are unavailable for small-area geographies. A prototype European Socio-economic Classification[38] has been developed, but comparable small-area data (from national censuses) for all areas are not yet available. Limited measures of housing tenure and car ownership are available, though these may also reflect different cultural patterns between countries rather than deprivation per se (eg, the different role that renting plays in the German housing market[39]). Some studies have also questioned whether car ownership is a good indicator of deprivation.[40 41] Measures of unemployment might also be challenged as not fully comparable either, due to the large-scale diversion of working-age adults into economic inactivity (eg, disability benefits) during the 1990s across many European countries.[42] Exploring options to overcome these methodological challenges might be a useful avenue for future research.

Data restrictions mean we were unable to explore systematically the degree of social segregation or migration within each region. Spatial inequalities observed could simply reflect greater population heterogeneity between districts within each region. However, evidence comparing WCS with the Ruhr and Nord-Pas-de-Calais does not support this hypothesis.[43 44] Nor can we say how spatial inequalities in LE changed within these regions over time, since the analysis is also confined to a single time period. Lack of individual-level data and common markers of socioeconomic status meant that this study was also confined to a focus on spatial differences in LE. If data had been available, analysis by inequalities by socioeconomic status or other characteristics (eg, rurality and ethnicity) may have led to different conclusions. For example, in Northern Moravia, the gap in male LE between districts was approximately 5 years,[45] but the gap in LE between the highest and least-educated men has been enumerated at 16.5 years.[46]

The more pronounced spatial inequalities in LE in three of the four UK regions, especially WCS, are notable. What factors might help account for this? As reported elsewhere, despite the relatively high levels of mean prosperity and lower unemployment, WCS and the other British regions have higher levels of relative poverty, income inequality and single person and lone parent households compared with postindustrial areas of mainland Europe.[32] There is also a more mixed pattern on some other indicators (eg, social capital and educational attainment).[32] It would be appropriate to consider the sociopolitical context to this. Others have contrasted the UK 'path destructive' road to deindustrialisation, characterised by the growth of a low-wage service sector and reduced social protection, with alternative strategies pursued in mainland Europe.[24 47] It has been argued that a more rapid adoption of neoliberal politics by local government in WCS alongside greater vulnerability to the deleterious impacts of associated economic policies might provide some basis for explaining the findings for WCS.[24 48]

There may be differences between regions in the homogeneity of the populations, and the degree to which there is social segregation. It is possible that the greater disparities observed in WCS could be due to greater social segregation rather than larger socioeconomic inequalities (although the likelihood of this is reduced by the same finding being observed at two different sizes of subregional districts). The limited analyses available (comparing spatial segregation in Nord-Pas-de-Calais and Merseyside with WCS) suggests that this cannot provide a wholly adequate explanation for the results shown here.[31] Nor is it clear that stronger within-region migration (from the unhealthiest to the healthiest districts) in WCS can explain these differences. One comparative study of WCS and the Ruhr (1995–2008) suggests that this pattern took place in both regions and, if anything, seemed to be slightly stronger in the Ruhr than in WCS.[43] This view is supported by Popham et al,[49] who argued that selective out-migration is not the only or most important reason for the wide levels of health inequality seen in the region.

Differences in overall population change might provide a partial explanation. Recent long-run analysis of commune-level data for France by Ghosn et al[50] found that population growth was associated with decreases in relative mortality. Between 1982 and 2005, while most of the regions included in our study saw little change in their population, WCS saw a marked decline; while Saxony saw an even larger loss of its population over a shorter time frame.[30] This might explain why inequalities in LE were wider in the Scottish region, but the much narrower inequalities in Saxony suggest that this may not be the whole story.

It may be that in other countries, 'protective' factors such as lower levels of income inequality (Northern Moravia),[51] higher levels of social capital (The Ruhr)[43] or fewer lone parent or single person households (Nord-Pas-de-Calais)[44] or a more managed deindustrialisation process, which included active labour market policies and re-employment in new industrial sectors,[24] might have partly mitigated against the health-damaging effects of deindustrialisation, reducing the extent of spatial inequalities in health. However, as yet unexplained region-specific factors are also likely to play a role. Within the UK, Swansea and South Wales have relatively narrow spatial inequalities in health and WCS has some of the widest. In the former case, this may partly reflect the more homogeneous social mix across ex-mining areas/ villages, compared with more metropolitan areas.

Difference in lifestyle factors (ie, worse health behaviours in WCS) could also play a role. This argument is more plausible for alcohol, since levels of consumption and alcohol-related harm are high in WCS for both genders compared with the other regions.[32] For smoking and diet, matters are less clear. Female smoking rates are higher in WCS compared with most regions but male smoking rates are similar across all regions.[32] Dietary indicators suggest that WCS compares

poorly with Nord-Pas-de-Calais but is very similar to Merseyside and Northern Ireland.[31] That said, any explanation based on health behaviours alone would be insufficient, as the underlying causes of these health behaviours would remain unexplained.

Finally, environmental factors, such as air pollution and climate, have also been proposed as possible explanations for health inequalities. Could these factors explain the results? Richardson *et al*[52] found that while exposure to particulate air pollution (PM10), and risk of some causes of mortality, was higher in low-income European regions, their mapping also revealed the concentration of the worst areas of pollution in East European regions (including Silesia and Northern Moravia). Although vitamin D deficiency (linked to lower levels of sunlight) may be higher in WCS than some other regions, the detrimental impacts on health are likely to be observed among older people.[53] Decomposition of the excess mortality observed in WCS compared with European regions shows it to be greatest among the working-age population, especially young men and middle-aged women.[30] It therefore seems less plausible that the observed difference in spatial inequalities can be attributed to environmental factors.

## CONCLUSIONS

Subregional spatial inequalities in LE in WCS are wide compared with other postindustrial European regions, even after accounting for differences in the population size of the subregional districts. These spatial inequalities are particularly profound for men. By contrast, within-region spatial inequalities in LE were relatively low in the German and Czech regions. These data generally show similar patterns to that for inequalities by individual educational attainment for the parent countries.[54] Outside the UK, wider determinants of health (such as income distribution, positive social capital and family networks) may have acted to protect health in postindustrial regions. Future research could explore the contribution of these wider determinants of health to reducing spatial inequalities in mortality, especially in WCS.

**Acknowledgements** This project would not have been possible without the cooperation and assistance of a number of individuals and organisations. In particular, the authors would like to thank the following for all their help in providing the required data: National Records for Scotland (NRS), formerly General Register Office for Scotland; Northern Ireland Statistics and Research Agency; Office for National Statistics and Public Health Intelligence Team, Information & Intelligence Services, Liverpool Primary Care Trust; Office for National Statistics (Vital Statistics); INSEE and Centre d'épidémiologie sur les causes médicales de décès (CepiDc; Original data provided by Observatoire Régional de la Santé (ORS), NPdC); SPMA (https://www.wiv-isp.be/epidemio/spma), Public Health and Surveillance, Scientific Institute of Public Health, Brussels, Belgium; NRW Institute of Health and Work (LIGA.NRW); Statistisches Landesamt des Freistaates Sachsen; Czech Statistical Office; Central Statistical Office of Poland—Local Data Bank. The authors would like to thank the reviewers, Peter Congdon and Paul Norman, for their useful comments on an earlier draft of this paper.

**Contributors** GM and DW conceived the idea for the paper and designed the study. All authors were involved in the acquisition of data, its analysis and interpretation. All authors contributed to the drafting and revision of the paper.

**Funding** This research received no specific grant from any funding agency in the public, commercial or not-for-profit sectors.

**Competing interests** None.

**Provenance and peer review** Not commissioned; externally peer reviewed.

**Data sharing statement** The data used to create figures 1 and 2 are available on request from the corresponding author.

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
