## [Reviewer comments · BMJ Open]

Some articles will have been accepted based in part or entirely on reviews undertaken for other BMJ Group journals. These will be reproduced where possible.

ARTICLE DETAILS

TITLE (PROVISIONAL)	Spatial inequalities in life expectancy within post-industrial regions of Europe: a cross-sectional observational study
AUTHORS	Taulbut, Martin; Walsh, David; McCartney, Gerry; Parcell, Sophie; Hartmann, Anja; Poirier, Gilles; Strniskova, Dana; Hanlon, Phil

VERSION 1 - REVIEW

REVIEWER	Peter Congdon QMUL, London, UK
REVIEW RETURNED	18-Mar-2014

GENERAL COMMENTS	Regrettably, I do not think this paper contains enough original material of substance to justify publication. The methods issues raised by the paper (e.g. measuring spatial inequality) are not discussed (e.g. there is no mention of slope index). The chosen measures (range, interquartile range) are not necessarily the best for the purpose. The analysis is quite limited with no attempt at regression, no mapping, no consideration of spatial clustering, etc. I am sure it would be possible to obtain some measures of area SES, rurality etc across the 160 Districts. Explanations provided for the results are not based on quantitative analysis (e.g. regression) but on speculation (e.g. in the Discussion). Explanations provided for any enduring effect on health of past industrial reliance or recent industrial decline were not convincing. A schematic discussion with postulated causal pathways would have been useful. Among issues not mentioned are selective out-migration (leaving concentrations of relatively high morbid populations) and environmental pollution. I also felt that the intra-regional heterogeneity was not sufficiently discussed or considered. For example, industrial activity in West Central Scotland was geographically concentrated within that region (e.g. Clydeside). There is acknowledgement of the standard literature (e.g. on MAUP) but no attempt to measure intra-regional diversity or similarity.
--

REVIEWER	Paul Norman School of Geography University of Leeds UK
REVIEW RETURNED	01-Apr-2014

GENERAL COMMENTS

This is an interesting and very useful paper with appropriate design and methods commensurate with the data availability and what this works aims to achieve. In the main it is very clear what the authors are doing and why and what they have found.

A few elements would benefit from a small amount of revision / reconsideration:

The opening paragraph could be widened out in terms of other work cited, partly because it could do with it anyway, but also because a recent paper (Tyner 2014) is rather underwhelmed by a lack of work similar to that presented here. Expanding the citations here (including perhaps a note, “despite a lack of research found by Tyner 2014 ...”) would therefore give this paragraph greater breadth of evidence. I suggest considering for inclusion, for Europe (Hoffman et al 2014), for France (and whether widening inequalities) (Ghosn et al 2012) for UK (and whether widening inequalities) Norman et al. (2012) and for England (and whether widening inequalities) Rees et al. 2003; Hacking et al 2011). The research in these papers uses a variety of spatial units. This would also widen the discussions, particularly in paragraphs on pages 13 and 14, by comparing with further work rather than being somewhat inward facing as the citations currently are.

The division into small and large areas is a good idea. I am not convinced this is ‘accounting for the population sizes’ as stated on p 9, line 14 (as we might in a weighted regression) but rather stratifying or differentiating, though I don’t feel strongly about this. Similarly, Table 1 would benefit from some edits. The last time I looked, the region of Nord-Pas-de-Calais wasn’t in the Nation State of UK! It would be clearer to have the table differentiated by the small and large districts grouped together and having equivalents for Northern Moravia as for WCS. Several times when reading the results I confused myself as to what I was trying to understand in this regard.

Also, a lot of work has gone into figures 1 and 2 but I don’t readily get what these graphs tell me that boxplots, perhaps ranked by median LE, wouldn’t (apart from the range). My eye is drawn to the angle of slope within each region but I am not sure that this is of relevance (and isn’t noted, that I could see) and in any case is affected by both the number of subdivisions in each region and by the graph axes which aren’t consistent. I can see how these graphs are constructed and the intention (I think) but wonder if the message isn’t diluted by the complication introduced.

Otherwise, well researched, well written.

Ghosn W, Kassie D, Jouglu E, Salem G, Rey G, and Rican S (2012) Trends in geographic mortality inequalities and their association with population changes in France, 1975–2006. *European Journal of Public Health* 23: 834–840.

Hacking et al (2011) Trends in mortality from 1965 to 2008 across the English north-south divide: comparative observational study *BMJ* ;342:d508 doi:10.1136/bmj.d508

Hoffman et al (2014) Social differences in avoidable mortality between small areas of 15 European cities: an ecological study. <http://www.ij-healthgeographics.com/content/13/1/8>

Rees P, Brown D, Norman P & Dorling D (2003) Are socioeconomic inequalities in mortality decreasing or increasing within some British

	regions? An observational study, 1990-98. Journal of Public Health Medicine. 25(3): 208-214 Tyner, J. A. (2014). Population geography II Mortality, premature death, and the ordering of life. Progress in Human Geography, 0309132514527037 (Sorry these references aren't all complete, not enough time to type out all the names etc!)
--	--

VERSION 1 – AUTHOR RESPONSE

Reviewer 1

Comments Responses

The methods issues are not discussed (there is no mention of the slope index). The chosen measures are not necessarily the best for the purpose.

The paper acknowledges that using these measures would allow for more robust analyses of spatial inequalities, using the slope index of inequality and relative slope index of inequality. However, to do this would require a ranking of areas by a consistent, international measure of socio-economic status – the difficulties in achieving this are outlined below.

There is a lack of measures of SES, rurality etc.

While a European measure of SES has been created, it is not yet available at small-area level. Other small-area measures of SES (such as car ownership, housing tenure and unemployment) are also discussed – these could be challenged as not comparable enough to provide a ranking method consistent between countries.

Among issues not mentioned are selective out-migration and environmental pollution.

These issues, together with climate, are now mentioned in the discussion. Intra-regional heterogeneity was not sufficiently discussed or considered.

This is now discussed – the limited evidence we have available (using dissimilarity index data) does not suggest WCS has a more spatially polarised population than Nord-Pas-de-Calais or Merseyside.

Reviewer 2

Comments Responses

Suggestions for broadening out the literature and discussion.

These useful and relevant papers have been added to the introduction and discussion, highlighting the relevance of this issue to population geography, the potential role of overall population change to spatial inequalities in mortality and that these findings may not apply outside of our sample of post-industrial regions.

VERSION 2 – REVIEW

REVIEWER	Paul Norman School of Geography, University of Leeds, UK
REVIEW RETURNED	11-May-2014

GENERAL COMMENTS	The authors have re-submitted a revised manuscript and I am pleased to see the literature usage broadened out somewhat in both the introduction and in later discussions. I think the acknowledgements about other methods and lack of comparable small area indicators across Europe are useful additions. I previously commented that I didn't find figures 1 and 2 easy to get the message from so am slightly disappointed not to see a defence / justification from the authors but don't think this gets in the way of publication. Maybe it's me!
--